# Native Microbes Amplify Native Seedling Establishment and Diversity While Inhibiting a Non-Native Grass

**DOI:** 10.3390/plants12051184

**Published:** 2023-03-06

**Authors:** Liz Koziol, Thomas P. McKenna, James D. Bever

**Affiliations:** Kansas Biological Survey, University of Kansas, 2101 Constant Ave, Lawrence, KS 66044, USA

**Keywords:** arbuscular mycorrhizal fungi, inoculation, invasion, germination, grasslands, restoration, succession, symbiosis

## Abstract

Although several studies have shown increased native plant establishment with native microbe soil amendments, few studies have investigated how microbes can alter seedling recruitment and establishment in the presence of a non-native competitor. In this study, the effect of microbial communities on seedling biomass and diversity was assessed by seeding pots with both native prairie seeds and a non-native grass that commonly invades US grassland restorations, *Setaria faberi*. Soil in the pots was inoculated with whole soil collections from ex-arable land, late successional arbuscular mycorrhizal (AM) fungi isolated from a nearby tallgrass prairie, with both prairie AM fungi and ex-arable whole soil, or with a sterile soil (control). We hypothesized (1) late successional plants would benefit from native AM fungi, (2) that non-native plants would outcompete native plants in ex-arable soils, and (3) early successional plants would be unresponsive to microbes. Overall, native plant abundance, late successional plant abundance, and total diversity were greatest in the native AM fungi+ ex-arable soil treatment. These increases led to decreased abundance of the non-native grass *S. faberi.* These results highlight the importance of late successional native microbes on native seed establishment and demonstrate that microbes can be harnessed to improve both plant community diversity and resistance to invasion during the nascent stages of restoration.

## 1. Introduction

Despite increasing knowledge of grassland restoration best practices [1,2], including developments in site preparation methods and seed sourcing [3,4], many seeded grassland species fail to establish, even decades after restoration [2,5,6,7]. As a result, land managers seed with high-diversity native seed mixtures [1], often at great expense, to little avail. More innovative and integrated methods are necessary to improve native plant establishment and to close the gap between seed mix diversity and established (realized) diversity in the field.

Evidence is accumulating that microbial communities can play a significant role in structuring plant communities and that soil microbial community structure should be considered in restoration efforts [8]. Not all microbes affect plants equally and plants of varying life-histories may respond differentially to specific microbial communities. For instance, late successional plants seem to be particularly sensitive to the presence of late successional microbes, otherwise known as old-growth or remnant microbes from undisturbed native systems [9,10]. Several field studies have found that late successional plant establishment is improved with co-application of late successional microbes relative to fields with only disturbance adapted microbes, such as those present in early successional soils and in ex-arable soils [11,12,13,14]. This improved establishment may be driven by the presence of late successional symbiotic arbuscular mycorrhizal (AM) fungi [9,13,15], which can increase germination [14], seedling survival [13], and competitive ability [16] of late successional plant species. A better understanding of the microbial contribution to plant community establishment—and the relative effects of microbes on plants with variable life-histories—is needed to better predict plant community dynamics in a restoration.

Integrating soil community structure and native plant species life histories into restoration planning may improve establishment as well as help practitioners overcome other restoration obstacles. One of the main hindrances to grassland restoration is the establishment of non-native and invasive plant species, which may form large monotypic stands and reduce plant diversity [17]. Weed control is often a substantial portion of a restoration/management effort, as land managers spend around 25% of their time utilizing different management techniques to reduce the abundance of undesirable species [1]. Efforts to increase native diversity through native seed addition in areas where non-natives have established dominance [18] and non-native removal followed by native seed addition [19] are often ineffective. However, studies have shown that when native plants establish well early (priority effects), non-native establishment may be inhibited [17,20]. If restoration practitioners can match native plant species with their compatible soil microbiome, there should be gains in initial germination and establishment of a diverse plant community and therefore the abundance of undesirable species should be reduced.

In this study, we assessed the impact of soil microbial community composition on the dynamics of native prairie plant community establishment in the presence of a non-native grass, *Setaria faberi* (foxtail), which often dominates in the early stages of prairie restoration throughout the Midwest. In the greenhouse, pots were seeded with a consistent density of early and late successional native plant species from multiple functional groups and the non-native *S. faberi*. Soil inoculation treatments included a sterile soil control (non-inoculated), whole soil from an ex-arable field, late successional AM fungi isolated from a nearby remnant tallgrass prairie, and a combination of late successional prairie AM fungi and ex-arable soils. We test the hypotheses that (1) late successional plants would benefit from late successional AM fungi, that (2) non-native plants would outcompete native plants in non-inoculated and ex-arable soils, that (3) early successional plants would perform the best with early successional microbes (ex-arable soils), or alternatively, that (4) early successional plants would be unresponsive to microbes.

## 2. Results

### 2.1. Plant Community Composition among Soil Inoculation Treatments

Non-metric multidimensional scaling (NMDS) showed a strong separation between non-inoculated and inoculated pots on Axis 1 and a separation of inoculation type with native AM fungi vs. ex-arable soil on Axis 2 (Figure 1). PERMANVOA results statistically supported effects of soil treatment (F_3,43_ = 5.827, *p* = 0.001) and block (F_10,43_ = 1.240, *p* = 0.001) on plant community composition. The pairwise PERMANOVA indicated that non-inoculated controls were significantly different from all inoculated soils (native AM fungi and native AM fungi + ex-arable soil both *p* = 0.006, ex-arable soil *p* = 0.012), and native AM fungi were significantly different from ex-arable soils (*p* = 0.012) and marginally different from native AM fungi + ex-arable soils (*p* = 0.090). There was weaker evidence that ex-arable soils were different from native AM fungi + ex-arable soil (*p* = 0.132). However, it should be noted that pairwise PERMANOVA analyses are not capable of including block as a predictor, which was a highly significant in the overall PERMANOVA.

### 2.2. Species-Specific and Diversity Responses to Soil Inoculation Treatments

Twenty-four intrinsic species were found to be contributing to the separation in plant community composition among the soil inoculation treatments (Table 1 and Appendix A, Figure 1). The seeded non-native grass *S. faberi* was associated with the sterile (non-inoculated) treatment. The early successional forb *Bidens artistosa* (*p* = 0.001) was associated with ex-arable soil inoculation (Table 1, Figure 1) and comprised more than 27% of the total pot biomass in ex-arable soil treatment. Other native early successional forbs (*Coreopsis tinctoria* and *Ratibida columnifera),* four non-native plants (*Melilotus officinalis, Cardamine hirsute, Medicado lupulina*, and *Viola sororia*), and three late successional forbs (*E. pallida, E. yuccifolium*, and *A. tuberosa*) were also associated with the ex-arable soil treatment. Many native early and late successional plants were associated with the native AM fungi + ex-arable soil treatment. Late successional plants associated with native AM fungi or native AM fungi + ex-arable soil inoculation include one legume (*Baptisia alba)*, two grasses (*Andropogon gerardii* and *Schizachrium scoparium)*, and several forbs (*Coreopsis lanceolata*, *Heliopsis helianthoides*, and *Scrophularia lanceolata)*.

Rank order abundance is presented for the 10 most abundant plant species in each soil inoculation type (Figure 2), excluding the seeded non-native species *S. faberi*, which was the most abundant plant in all treatments (discussed below). Rank abundance indicated that native AM fungi + ex-arable soil had both the most late successional plants (6) and the fewest non-native species (0) of any inoculation treatment, while pots inoculated with ex-arable soil were dominated by early successional and non-native plants (7) and had the fewest late successional plants (3) (Figure 2). Overall plant community diversity was significantly different among the soil inoculation treatments (*p* = 0.01; Table 2A), where plant community diversity was 33.5 % greater with the native AM fungi + ex-arable soil relative to ex-arable soil inoculations alone (Figure 3A). Given that total abundance was not different among soil inoculation treatments (Figure 3B, Table 2A, *p* = 0.3), these changes in plant community composition were driven by changes in the dominant plant species with different soil inoculation treatments.

### 2.3. Inoculation Driven Shifts in Invasion, Successional Processes, and Functional Group Representation

Inoculation treatments strongly affected the proportion of native/non-native plant species in each community (Table 2A, Figure 4A, *p* < 0.0001) with the greatest native proportion (59%) and the lowest non-native proposition (41%) being in pots inoculated with both native AM fungi + ex-arable soil. Categorical predictors of species composition highlighted that native abundance was significantly increased with inoculation (Table 2A, 67.6%, *p* < 0.0001), while non-native abundance was significantly reduced with inoculation (Table 2A, 15.7%, *p* = 0.0002). Non-native abundance significantly differed among the various inoculation types (Table 2A, *p* = 0.009), where pots inoculated with both native AM fungi + ex-arable soil had the least non-native abundance (25.8% less than controls) and had the greatest native seeded abundance (Figure 4B, 74.7% more than controls).

Inoculation and inoculation source also affected the abundance of different plant successional stages and functional groups among native plants (Table 2B). Early and late successional species benefited from the presence of native AM fungi (Table 2B, both *p* < 0.001). Late successional abundance was greatest with native AM fungi + ex-arable soil microbes (Figure 4C), with 19% more late successional plant biomass than the ex-arable inoculated pots. Forbs (*p* = 0.01), grasses, (*p* = 0.052), and legumes (*p* = 0.003) responded differently to inoculum composition. Legume abundance was 308% greater with native AM fungal inoculation than without (Table 2B, Figure 5A, *p* < 0.0001). Native grasses were most abundant with native AM fungi + ex-arable soil inoculation, growing 40.2% larger than when in ex-arable soils and 327% larger than controls (Figure 5B). Native forbs were most abundant in the whole soil inoculations treatments with 72.5% more forbs than controls and 28% more forbs than the other inoculation treatments (Figure 5C).

While the above presented data focused on total seedling abundance, it should be noted that the main effect of inoculation had consistent significant effects across the first and second harvests for most metrics including native abundance, non-native abundance, early successional abundance, native forb abundance, native grass abundance, and total diversity (Appendix A, Appendix A). The main effect of inoculation on total abundance became insignificant over time. Important differences between harvests were that native legume abundance and relative invasion became increasingly significant over time (both *p* < 0.0001), especially with native AM fungi inoculation (Appendix A, both *p* < 0.0001). Additionally, inoculation improved late successional plant biomass the first harvest (Appendix A, F = 8.06, *p* = 0.008) but not the second. The second harvest pattern was driven by late successional plants not benefiting from controls or ex-arable soil while largely benefiting from native AM fungi (Appendix A F = 3.63, *p* = 0.067).

## 3. Discussion

Predicting plant community establishment in grassland restorations is highly challenging across the globe [2,21,22] and finding restoration strategies that boost native plants while reducing non-native establishment is necessary to optimize restoration practices [1]. Recent evidence suggests that microbes—especially native microbes—may play a direct role in seedling recruitment and establishment and may differentially affect plants from different life-histories [16,23]. Here, the microbial contribution to seedling growth and establishment in a prairie community was assessed by seeding pots with a consistent density of native and non-native plants and applying different microbial communities. Among nearly every metric assessed in this study, plant community composition was improved with the addition of native arbuscular mycorrhizal (AM) fungi isolated from remnant prairie. Overall, the native AM fungi + ex-arable soil treatment conferred the greatest abundance of native and late successional seedlings, total diversity, native legume, and grass abundance, while non-native plant competition was reduced. These results largely agree with a growing body of research observing that native plant diversity and establishment can increase while non-native plant abundance and diversity can decrease in the presence of native AM fungi [11,12,13,14,24,25], including a paired field study using the same microbial amendments in the field [26,27]. Together these results highlight the importance of microbes for native seed germination and establishment, and demonstrate that microbial mediated improvement in native establishment can contribute to improved plant community diversity and resistance to invasion in restoration efforts.

In this study, it was observed that early successional plants benefited from most inoculation treatments similarly, counter to the hypothesis that early successional species would perform best with early successional inocula. The one notable exception is the early successional forb *Bidens aristosa*, which was highly abundant in ex-arable soils (>25% of pot mass), more so than the native AM fungi + ex-arable soil treatment. *B. aristosa* is a species of low conservation concern that readily establishes in restoration [28,29]. Overall, we found stronger support for the hypothesis that early successional plants are less dependent on or sensitive to microbial composition, more closely matching previous work assessing mycorrhizal response across a successional gradient [10,30,31].

From the perspective of restoration in situ, the question is whether adding later successional native mycorrhizal fungi collected from an undisturbed system would improve seedling establishment of native late successional plants. Because late successional plants are so strongly dependent on late successional microbes in both the field and greenhouse assays [11,13,14], we expected that late successional seedlings would benefit from late successional AM fungal additions. In general, this study confirmed the findings of past work, as seedlings of late successional species, as a guild, were strongly affected by inoculation during the first harvest and grew largest with native AM fungi or native AM fungi + ex-arable soils overall. However, certain late successional seedlings demonstrated stronger responses to native AM fungi than others. Intrinsic species analysis indicated that there were three late successional forbs associated with ex-arable whole soils, confirming past studies showing that a few late successional plants can establish well in restoration regardless of microbial inoculation [13]. Yet, most late successional species regardless of functional group were associated with the native AM fungi + ex-arable soil inoculations. This pattern was similar to responses observed in the rank abundance analyses, where late successional grasses and legumes were found to be especially sensitive to late successional AM fungal additions. Past work with late successional legumes has also found that their germination and growth is highly dependent on the presence and inoculation of late successional microbes [12,26,32,33]. Overall, these data support the hypothesis that adding late successional native mycorrhizal fungi collected from an undisturbed system can improve native and late successional seedling establishment.

Lastly, we predicted that non-native plants would outcompete native plants in ex-arable soils but not with native AM fungal additions, given both that non-native plants can be hindered by late successional microbes [16,24] and that non-native plants are highly effective at establishing in restorations in ex-arable soils despite native seed addition [19]. In this study, non-native plants outcompeted native plants in the non-inoculated control, with ex-arable soil inoculations and with AM fungi alone. However, the addition of native AM fungi to ex-arable soil reversed this pattern and native plants outcompeted the non-native plants. Although the non-native *S. faberi* seeded into this study is mycorrhizal [9], it was directly or indirectly inhibited after inoculation with native AM fungi + ex-arable soil. Past restoration experiments have shown that a diversity of non-native plant species can be inhibited by a suite of late successional microbes in the field [12,16,24], especially late successional AM fungal inoculations [26,27]. Thus, we conclude that late successional microbes—especially late successional AM fungi—can be used as a tool to reduce non-native plants in grassland restoration. Tests of native AM fungal inoculation effects should be conducted on a wider variety of non-native plants that commonly invade restoration to further inform whether microbial additions can be used to optimize restoration outcomes by promoting natives and inhibiting non-native plants in plant communities worldwide.

## 4. Conclusions

These data agree with the commonly observed patterns in conventional restorations with no microbial amendments; where plant community diversity is generally low, both early successional and non-native plants dominate, and few late successional plants germinate and establish [5,8]. However, these results indicate that adding native AM fungi to restoration seedling practices can increase diversity by promoting late successional seedling establishment while inhibiting overly competitive non-native plants. Although laboratory cultured AM fungi were used in this study, similar patterns have been found using native late successional whole soil from a reference donor site [12,14,16,34,35]. Given this work and others on the microbial contribution to plant community establishment and succession in restoration, future restoration practices with a goal of increasing native plant community diversity or inhibiting non-native plants should also consider amending soils with late successional microbes including native AM fungi at restoration initiation, where seedling native and non-native dynamics may determine restoration outcomes for many years [17,18,19,20].

## 5. Materials and Methods

### 5.1. Methods

Pots (6 L) were partially filled (70%) with a steam sterilized (twice at 77 °C) 50:50 sand:soil mixture (15.8 ppm P via Melich extraction, 26.55 ppm NO_3_-N and 5.8 ppm NH_4_-N via KCl extractions). One of four soil inocula was added (200 cm^3^ total), pots were filled the rest of the way with the sterile background sand:soil mixture, and then planted with the same seed mix. Pots were arranged in a randomized complete block design with 11 replicates (1 planting treatment × 4 soil inocula × 11 replicates = 44 pots total). Pots were well-watered twice daily for 2 min with a 2 L per hour emitter via drip irrigation to prevent cross contamination of soil inoculation treatments.

### 5.2. Inoculum & Seed Collection

Pots were inoculated with one of four soil treatments: (1) live ex-arable soil, (2) live late successional prairie AM fungi, (3) live ex-arable soil and live late successional prairie AM fungi, or (4) sterilized soil (non-inoculated controls). The ex-arable whole soil (henceforth termed “ex-arable soil”) inoculum was collected in 2017 at The Land Institute’s Perennial Agriculture Project Field Station located in Lawrence, KS, USA (39.001311°, −95.320337°). The site was dominated by *Bromus inermis* (smooth brome) that was planted at least 20 years prior to this experiment. The prairie AM fungi inoculum was created using single species fungal cultures that had been isolated based on spore morphology. The spores for cultures were isolated from an unploughed remnant native prairie in Lawrence, Kansas (39.04619208°, −95.2050294°). Cultures were grown in 2016 for one year in a sterilized sand:soil mixture (10.15 P ppm via Melich extraction, 7.375 NO_3_-N ppm and 22.2 NH_3_-N ppm via KCl extractions) prior to use in this experiment. A native AM fungal community mixture was created by mixing 7 AM fungal species: *Scutellospora dipurpurescens, Gigaspora gigantea*, *Funneliformis mosseae*, *Funneliformis geosporum*, *Glomus mortonii*, *Rhizophagus diaphanous*, and *Claroideoglomus claroideum*. Past work has shown that these native AM fungal species benefit native prairie plants from this region [27,36]. AM fungal spore density was approximately 30 spores/cm3 or 25,132 spores/kilogram. This late successional prairie AM fungal mixture is henceforth termed “native AM fungi.”

Native plant seeds were obtained from Hamilton Native Outpost (Elk Creek, MO, USA) and Prairie Moon Nursery (Winona, MN, USA) (Appendix A). For each of the 43 plant species, the weight of an estimated 200 seed per each native species was evenly divided into 44 bags, where each bag represented one pot. For the non-native plant species addition, seeds of *Setaria faberi* (commonly referred to as “foxtail”) were collected from a nearby prairie restoration experiment, where it can be abundant [27]. Twenty seeds of *S. faberi* were added to each bag. All seeds were then cold (4 °C) moist stratified in damp sterile sand for 3 months. In late May of 2017, the seed mixture was hand broadcast onto the top of each pot and raked in with a sterile gloved hand. Several non-seeded species germinated Appendix A.

### 5.3. Data Collection

Pots were harvested in early July and again in early September of 2017 by cutting the aboveground biomass down to one inch above the soil surface. Plants were sorted by species, dried (60 °C for a minimum of 48 h), and weighed. Each plant species was identified as native or non-native, and as a grass (*Poaceae* or *Gramineae*), forb (herbaceous plant not in *Fabaceae, Poaceae* or *Gramineae*), or legume (*Fabaceae*). Seeded native plants were identified as being early successional or late successional based on a local coefficient of conservatism score, where 1–4 were identified as early successional and 5–10 were identified as late successional [29]. Total plant biomass (both harvests combined) is reported in the manuscript, while individual harvests can be found in the Appendix A.

### 5.4. Statistical Analyses

For multivariate plant community analysis, a relative abundance matrix based on the aboveground biomass of each species within a pot was used. The effects of soil inoculation treatments on plant community establishment were visually assessed using Nonmetric multidimensional scaling (metaMDS function; vegan package in R statistical software in R [37]) and plant species driving the observed pattern (intrinsic species) were identified (envfit function). Permutational MANOVAs (adonis2 function; vegan package in R) were used to statistical test for differences in plant communities among the soil inoculation treatments. Pairwise comparisons among soil inoculation treatments were performed with the RVAideMemoire package [38] in R (pairwise.permanova function with Bonferroni adjusted *p* values).

The abundance (transformed aboveground biomass (ln(1 + biomass)) and inverse Simpson’s diversity index (vegan package in R) of establishing plants were analyzed using SAS statistical software (proc mixed; [39]). Separate analyses were run for native, non-native, and total (native + non-native) plant species. Data are presented for the combined harvests and closely matches patterns observed in individual harvests as reviewed at the end of the results and in the Appendix A. Due to sample uncertainty between two non-inoculated pots during harvest, two non-inoculated controls were removed from analysis for the first harvest. To aid in the visualization of plant community composition among inoculation treatments, abundance data were averaged by plant species within inoculation category (native AM fungi, ex-arable soil + native AM fungi, ex-arable soil or non-inoculated) and then arranged by rank abundance for each inoculation category. These data are presented for the most abundant 10 species after *S. faberi* (which is depicted in Figure 4).

## Figures and Tables

**Figure 1 plants-12-01184-f001:**
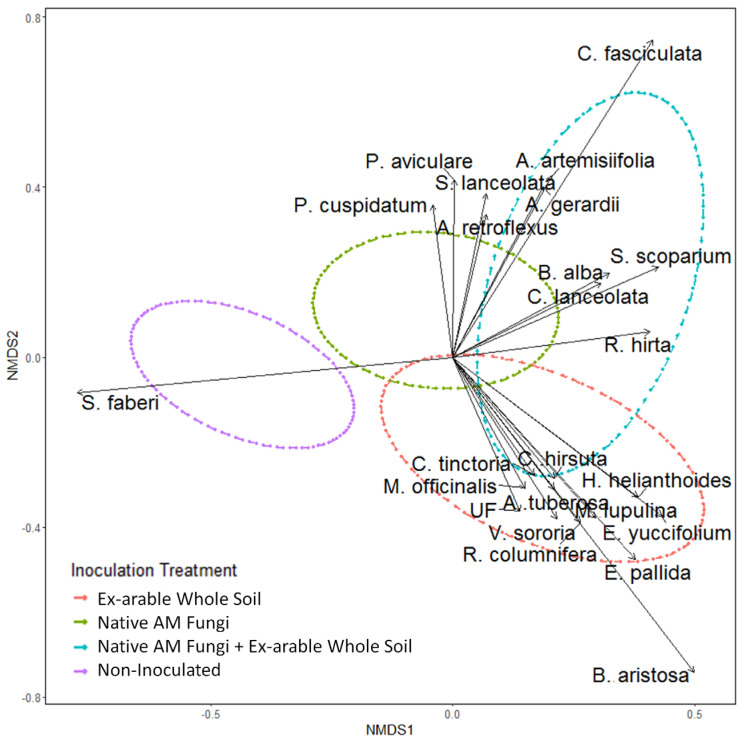
NMDS ordination of plant community composition among the four soil inoculation treatments (indicated by ellipse and color). The vectors are plant species contributing (*p* < 0.10 from intrinsic species analysis; Table 1) to the differences in plant communities among inoculation treatments.

**Figure 2 plants-12-01184-f002:**
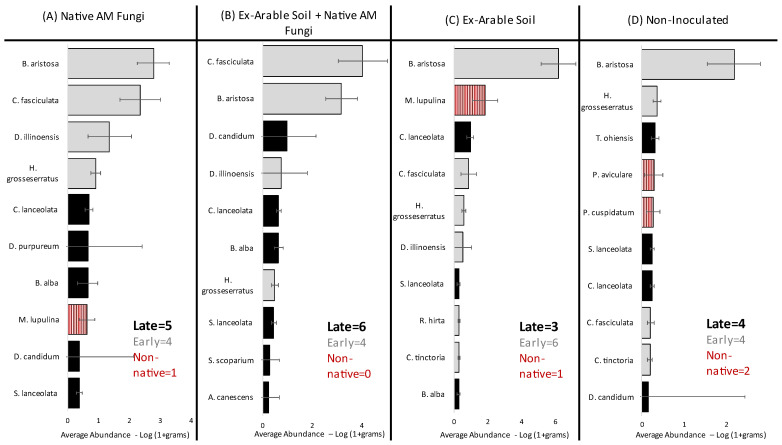
Rank abundance for pots inoculated with (**A**) native AM fungi, (**B**) native AM fungi + ex-arable soil, (**C**) ex-arable soil, or (**D**) non-inoculated control. Bars represent average species abundance (g), and error bars are standard error. Bar color is determined by plant coefficient of conservatism (CC) score, where gray bars represent early successional plants (CC = 1–4), black bars represent late successional plants (CC = 5–10), and red striped bars represent non-native (CC = 0) and non-seeded species.

**Figure 3 plants-12-01184-f003:**
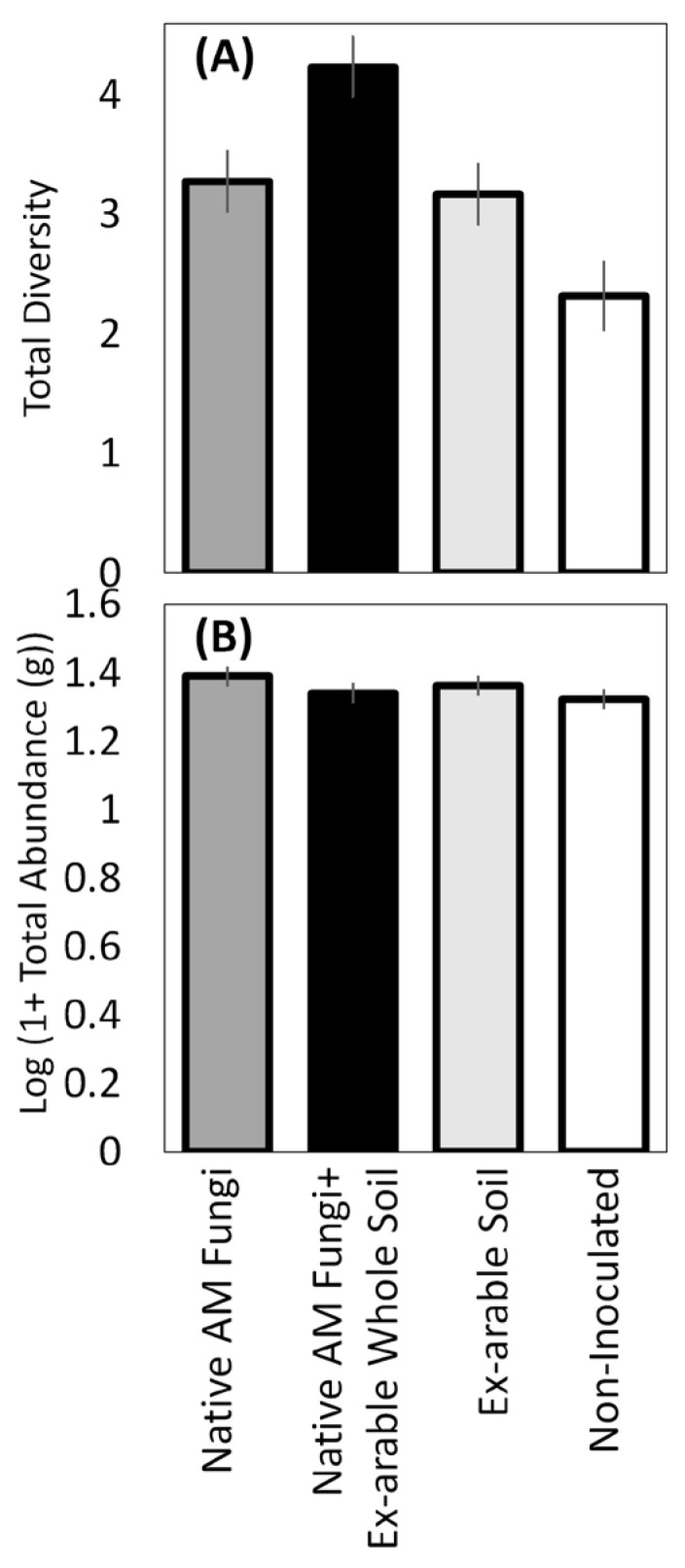
The effect of inoculation treatment (native AM fungi (dark grey), native AM fungi + ex-arable soil (black), ex-arable soil (light grey), and the non-inoculated control (white)) on (**A**) total plant community diversity (inverse Simpson’s index) and (**B**) total seedling biomass. Bars represent the LS means of plant aboveground biomass and error bars are standard error from the Proc GLM models.

**Figure 4 plants-12-01184-f004:**
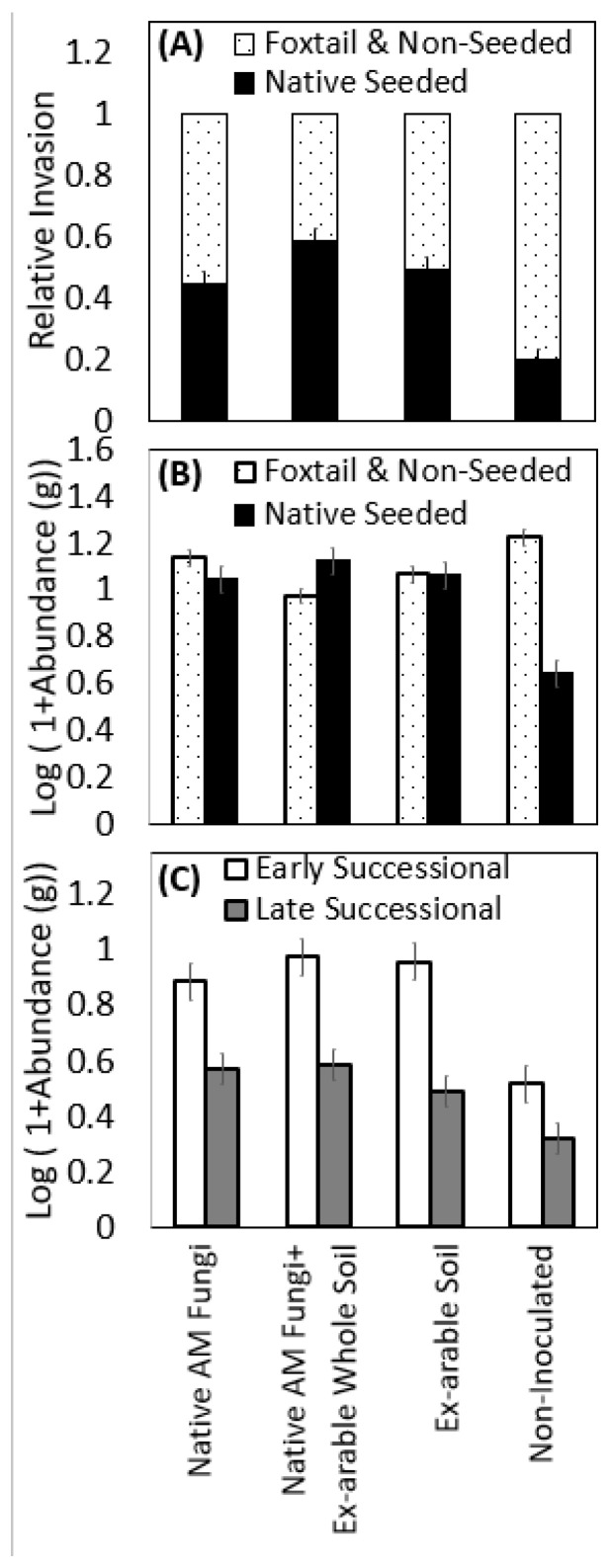
The effect of inoculation treatment on (**A**) the relative invasibility (proportion of native vs. non-native plants) and (**B**) the abundance (Log (1 + aboveground biomass)) of native (black bars) versus non-native plants (*S. faberi* (foxtail) and non-seeded; white patterned bars), and (**C**) the abundance of seeded early successional (white bars) and late successional plants (grey bars). Bars represent the LS means of plant growth and error bars are standard error from the Proc GLM models.

**Figure 5 plants-12-01184-f005:**
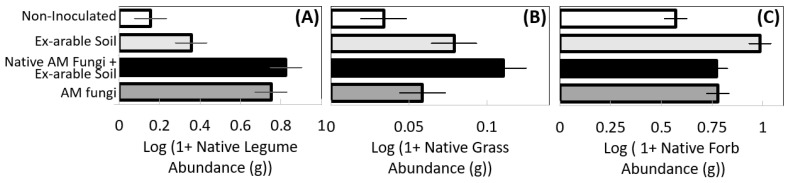
The effect of inoculation treatment (native AM fungi (dark grey), native AM fungi + ex-arable soil (black), ex-arable soil (light grey), and the non-inoculated control (white)) on native (**A**) legume, (**B**) grass, and (**C**) forb abundance. Bars represent the LS means of plant aboveground biomass and error bars are standard error from the Proc GLM models.

**Table 1 plants-12-01184-t001:** NMDS axes scores and *p*-values of the plant species contributing (*p* < 0.10) to the separation in plant community composition among inoculation treatments.

Species	NMDS1	NMDS2	*p*-Value
*B. aristosa*	0.4978811	−0.741185	0.001
*E. pallida*	0.3758625	−0.475515	0.001
*C. fasciculata*	0.4120677	0.7460587	0.001
*S. faberi*	−0.77588	−0.083553	0.001
*E. yuccifolium*	0.430899	−0.373672	0.002
*H. helianthoides*	0.3823069	−0.329738	0.003
*R. columnifera*	0.2631774	−0.388539	0.004
*M. lupulina*	0.2941195	−0.378259	0.004
*A. artemisiifolia*	0.2044813	0.4260718	0.005
*S. scoparium*	0.4244076	0.2123334	0.009
*P. aviculare*	0.0028758	0.4169241	0.016
*V. sororia*	0.2141607	−0.380134	0.018
*R. hirta*	0.4068963	0.0594642	0.024
*UF (Unknown Forb)*	0.1385269	−0.362878	0.03
*B. alba*	0.323966	0.1972003	0.035
*S. lanceolata*	0.0683162	0.3843318	0.036
*A. tuberosa*	0.2083887	−0.310465	0.045
*P. cuspidatum*	−0.042617	0.3576915	0.048
*A. gerardii*	0.1858767	0.4012776	0.049
*C. lanceolata*	0.3055674	0.1741912	0.071
*A. retroflexus*	0.0682149	0.3365135	0.076
*C. hirsuta*	0.2094112	−0.284899	0.08
*C. tinctoria*	0.1679516	−0.278248	0.09
*M. officinalis*	0.1491952	−0.308403	0.097

**Table 2 plants-12-01184-t002:** The main effect of inoculation treatment (bold) and contrasts (italics) results from the PROC GLM model in SAS on (**A**) total diversity, abundance, and nativeness and (**B**) for seedling abundance within successional stages and functional groups.

(A)		Total Diversity		Total Abundance		Native Abundance		Non-Native Abundance		Relative Invasion	
Predictors	DF	F Value	*p* Value	F Value	*p* Value	F Value	*p* Value	F Value	*p* Value	F Value	*p* Value
**Inoculation Treatment**	3	8.62	0.0003	1.2	0.326	16.43	<0.0001	9.41	0.0002	19.72	<0.0001
**Block**	10	1.58	0.165	2.13	0.054	2.43	0.0292	0.65	0.7586	1.66	0.1367
**Contrasts**											
*Inoculated vs. Non-* *Inoculated*	1	15	0.0006	1.85	0.184	48.17	<0.0001	17.26	0.0002	51.92	<0.0001
*Inoculated with Native AM Fungi vs. Not*	1	15.08	0.0006	0.75	0.393	18.23	0.0002	7.16	0.0119	21.01	<0.0001
*Ex-arable Soil vs. Native AM Fungi + Ex-Arable Soil*	1	8.95	0.0057	0.33	0.569	0.61	0.4409	3.77	0.0616	3.13	0.0871
*Differences Among Live Inocula*	2	5.43	0.0102	0.88	0.426	0.56	0.5785	5.48	0.0093	3.62	0.0391
**(B)**		**Early** **Successional Plants**		**Late** **Successional Plants**		**Native Forbs**		**Native Grasses**		**Native Legumes**	
**Predictors**	**DF**	**F Value**	** *p* ** **Value**	**F Value**	** *p* ** **Value**	**F Value**	** *p* ** **Value**	**F Value**	** *p* ** **Value**	**F Value**	** *p* ** **Value**
**Inoculation Treatment**	3	10.52	<0.0001	4.93	0.007	9.87	0.0001	5.03	0.0061	16.88	<0.0001
**Block**	10	2.41	0.0305	1.15	0.359	3.23	0.0062	2.57	0.0221	0.94	0.5102
**Contrasts**											
*Inoculated vs. Non-* *Inoculated*	1	30.52	<0.0001	13.04	0.001	19.46	0.0001	8.55	0.0065	29.62	<0.0001
*Inoculated with Native AM Fungi vs. Not*	1	8.52	0.0066	9.9	0.004	0	0.959	3.82	0.0601	46.89	<0.0001
*Ex-arable Soil vs. Native AM Fungi +Ex-Arable Soil*	1	0.04	0.8472	1.51	0.229	7.82	0.0089	2.4	0.1315	18.16	0.0002
*Differences Among Live Inocula*	2	0.52	0.6018	0.87	0.431	5.07	0.0127	3.27	0.0518	10.52	0.0003

## Data Availability

The data presented in this study are available in Appendix A.

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
