# Peer review of "Native Microbes Amplify Native Seedling Establishment and Diversity While Inhibiting a Non-Native Grass"

_plants, 2023, doi:10.3390/plants12051184_

Round 1

Reviewer 1 Report

This well-executed and important study with clear hypotheses builds on a body of work by the authors examining the effects of soil microbes on early and late-successional plants in grasslands. This study uses a greenhouse microcosm experiment to test whether AMF inoculation improves plant diversity, increases abundance of late-successional species, and reduces non-native abundance. This paper replicates – in a controlled setting – how restorationists apply seed additions and thus has important implications for grassland restoration. AMF inoculation increased native diversity, desirable late-successional species, and reduced the abundance of non-native species. This manuscript is generally well-written and structured. I have only a few suggestions for improving the clarity of the text and results.

Throughout the manuscript, you mention that one of the goals of the study is examine soil microbial effects on seed germination and you report that seed germination responded to soil inoculation treatments. However, although you sowed seeds at a constant density, germination per se was not quantified in this study and you cannot infer germination rates from data on biomass, which is the only response variable reported. Your design did not disentangle treatment effects on germination (adjusted for viability), seedling survival, and seedling growth. I recommend removing references to germination throughout the manuscript and focus on seedling biomass, which is main response variable presented.  

Lines 83-84: Which treatment is the “whole-soils” referring to? Is this combining the ex-arable treatment and AMF+ex-arable soil treatment together in the contrast? Also, the text states that AMF is significantly different from ex-arable soils but the p-value is 0.09. The methods do not mention a threshold for defining significance. It may be better to say “weaker effect” or something along those lines.

Lines 102-105: Your hypothesis predicts that late-successional species should be associated with Native AMF rather than ex-arable soil so why not indicate that late-successional species responded differently to soil treatments rather than lumping the responses of late-successional species to AMF+ex-arable and ex-arable treatments together? I was struck by the result that several late-successional species including E. pallida, E. yuccifolium, and A. tuberosa have positive NMDS 1 scores and negative NMDS 2 scores, like B. aristosa, and are associated with ex-arable soil, which differs from the expected response. Interspecific variation in the responses of late-successional forbs to AMF inoculation is a noteworthy pattern in the NMDS that gets little attention in the rest of manuscript.  

Figure 3: Is Simpson’s diversity being reported for total plant community diversity? Clarify which metric you are reporting here and in the text. Also, see comment below about richness.

Figure 5c: Did you consider dividing forbs into early- and late-successional since B. aristosa (early-successional) has such strong response to ex-arable soils? It seems like might be another angle to your hypothesis and provide greater relevance to the study if you find late-successional forbs are more responsive to AMF inoculation since forb enrichment is a key goal in grassland restoration.

Line 220: As a guild, late-successional forbs responded strongly, but within the guild the NMDS shows substantial variation among different species of late-successional forbs.   

Line 310: By focusing only on aboveground biomass, is it possible that the comparison between early- and late-successional species would differ if whole-plant biomass was examined? Resource allocation theory predicts that slower-growing, late-successional forbs should preferentially allocate resources below- versus above-ground whereas the opposite is expected for fast-growing, early-successional forbs. Perhaps your other studies shed light on how soil inoculation effects resource allocation.

Line 319: I don’t see where richness is reported in the Results or Supplemental material.

Author Response

Please see the attached document, where we respond to each line individually. 

Reviewer 2 Report

Title:

Native microbes help native seedling germination, establishment, and diversity while inhibiting a non-native grass

 The experiment is well performed and the theme is, short, novel, and very interesting, but minor corrections are needed.

 Abstract:

- Line 9: change "may" to "can"

- Line 10: change "on seedling …." to "in seedling ….."

- Line 12: change "restorations" to "restoration"

- Lines 15-16: delete "that", each "will" may change to "would", and add "and" before 3). The revised sentence "We hypothesized that 1) late successional plants would benefit from native AM fungi, 2) non-native plants would outcompete native plants in ex-arable soils, and 3) early successional plants would be unresponsive to microbes.".

Introduction:

- Line 26: delete "an". Preferably write "Despite knowledge of best practices for grassland restoration has increased, ….."

- Lines 70-74: You may correct the sentence as done in lines 15-16 in the abstract.

 Discussion:

- Line 194 (as well as line 13 in abstract): Write the full name of the AM fungi at the first time it is mentioned.

- Lines 194-196: write "Overall, AM fungi + ex-arable soil treatment conferred the greatest abundance of native and late successional seedlings, total diversity, native legume, and grass abundance, while non-native plant competition was reduced.".

 Materials & Methods:

- Line 260: the word "pots" is repeated. Delete one of them.

- Line 261: subscript "3" and "4" in NO3-N and NH4-N.

- Lines 262 and 265: change "inoculum" to "inocula or inoculums".

- Line 266: adhere to the journal's guidelines for "hour and minutes", such as h and min.

- Lines 279-280: the same comment of line 261.

- Line 293: it is °C

 Conclusion:

- Since there is no conclusion, it is preferable to add a conclusion of 3-4 sentences.

Author Response

Please see the attached document, where we respond to each comment individually. 

Reviewer 3 Report

I recommend the publication of the article because the scientific experimentation concerning studies on the activity of microorganisms and mycorrhizae on germination, plant growth and in particular on fungal-plant interaction is really interesting and topical. Studies on the differences between the use of indigenous and externally applied microorganisms are of great interest, especially for improving applications in agriculture.

The aim and objectives of the article have been stated and are very fascinating.

The use of native microorganisms for plant growth and defence is an important topic especially in organic farming with regard to the reduction of synthetic products and chemical fertilisers.  The work is certainly of international interest and the format applied is certainly suitable for a research paper. The work is original, of particular interest and can certainly stimulate research on this topic. The length of the article is good for the journal and the graphs and tables are clear and easy to understand. The conclusion summarises the aims of the work and future prospects.

Author Response

Please see the attached document where we responded to each reviewer comment individually. 
